# Optical and Physicochemical Characterizations of a Cellulosic/CdSe-QDs@S-DAB_5_ Film

**DOI:** 10.3390/nano12030484

**Published:** 2022-01-29

**Authors:** Manuel Algarra, Ana L. Cuevas, Ma Valle Martínez de Yuso, Rocío Romero, Beatriz Alonso, Carmen M. Casado, Juana Benavente

**Affiliations:** 1INAMAT2-Institute for Advanced Materials and Mathematics, Departamento de Ciencias, Universidad Pública de Navarra, Campus de Arrosadía, 31006 Pamplona, Spain; 2Unidad de Nanotecnología, Centro de Supercomputación y Bioinnovación, Servicios Centrales de Investigación, Universidad de Málaga, 29071 Málaga, Spain; analaura.cuevas@uma.es (A.L.C.); rociorp@uma.es (R.R.); 3X-ray Photoelectron Spectroscopy Lab., Central Service to Support Research Building (SCAI), Universidad de Málaga, 29071 Málaga, Spain; mvyuso@uma.es; 4Departamento de Química Inorgánica, Universidad Autónoma de Madrid, Cantoblanco, 28049 Madrid, Spain; beatriz.alonso@uam.es (B.A.); carmenm.casado@uam.es (C.M.C.); 5Departamento de Física Aplicada I, Facultad de Ciencias, Universidad de Málaga, 29071 Málaga, Spain

**Keywords:** CdSe quantum dots, dendrimers, regenerated cellulose modified film, optical properties

## Abstract

CdSe quantum dots nanoparticles were coated with the thiolated (DiAminoButane based dendrimer) DAB dendrimer of fifth generation (S-DAB_5_) and embedded in a highly hydrophilic regenerated cellulose (RC) film by simple dip-coating method (immersion in QD-dendrimer aqueous solution) as a way to get a flexible nano-engineered film (RC-4/CdSe-QDs@S-DAB_5_) with high transparency and photoluminescence properties for different applications. Optical changes in the RC film associated with QDs inclusion were determined by spectroscopic ellipsometry (SE) measurements, which provide information on changes caused in the refraction index and the extinction coefficients of the film, as well as by light transmittance/reflectance curves and photoluminescence (PL) spectra. Impedance spectroscopy (IS) and other typical physicochemical techniques for material characterization (TEM, SEM and XPS) have also been used in order to have more complete information on film characteristics. A comparison of RC-4/CdSe-QDs@S-DAB_5_ film optical characteristics with those exhibited by other RC-modified films depending on the type of dendrimer was also carried out.

## 1. Introduction

Colloidal quantum dot (QD) nanoparticles are semiconductor nanocrystals, with a core-shell structure and a diameter ranging ideally between 2 nm and 10 nm, which display unique electronic and optical properties (between bulk semiconductors and discrete molecules) based on both size and chemical composition [1,2,3,4,5]. QDs can emit light at wavelengths ranging from the UV to the IR, a photon emission at a longer wavelength than the one absorbed (electron-hole recombination process) [6]. Their properties include highest photostability, high extinction coefficient and brightness, magnetic, thermal and antibacterial characteristics as well as small size, this latter being a great advantage over other nanoparticles (NPs) used for multifunctional probes (polymeric or silica NPs), due to their large surface area [7]. 

In fact, QDs are nowadays of significant interest in different nanotechnology fields such as biomedical, electronics and optoelectronic devices due to their pure color emission, wide color gamut and high quantum efficiency [8]. Moreover, QDs also exhibit narrow band gap, sharp emission, and excellent spectral properties [9,10,11], but they are unprotected from external agents (oxidation, water, heat, or harsh environments) which significantly limit their long-term stability [12,13,14]. In this context, the inclusion of QDs into transparent polymeric matrices has been considered as a way of reducing fluorescence quenching and improving QDs stability [15]. Different QDs (CdSe, CdTe, CdSe/ZnS, carbon, nitrogen-doped carbon, graphene) and support matrices (poly(methyl methacrylate), polyamide, transparent wood, cellulose, etc.) for electronic devices, solar concentrators, white light-emitting diodes, sensors or agriculture (sunlight conversion films) have already been reported [16,17,18,19,20,21,22,23,24,25]. Among them, regenerated cellulose (RC), a flexible, transparent, biodegradable and low-cost material, exhibits appropriated characteristics for QDs modification [26,27,28,29]. Moreover, the high hydrophilic character of RC films has already permitted the inclusion of different nanoparticles by depth-coating in water solutions of desired NPs, a method that does not require complex processes [30,31,32]. In this regard, QDs coverage with dendrimers, such as thiol DAB, seems to increase photoluminescence intensity, reduce their possible toxicity and favor their link with the cellulose chains, as it was already obtained in our previous works [33,34,35]. 

Dendrimers are macromolecules synthesized with very high precision (monodispersed macromolecules) following a bottom-up approach, which allows an almost total control in both size and surface properties, making them an attractive platform for very diverse applications such as imaging or detecting agents (dye molecule), targeting components, radioligands or pharmaceutically active compounds [36,37]. In this context, it is known that polyamidoamine dendrimers (PAMAMs) can adsorb small molecules in a non-covalent way, which helps to the controlled delivery of fluorescent systems [38]. In fact, the binding of dendrimers to nanoparticles, more specifically to QDs, seems to preserve them from environmental effects, improving their emissive properties both at energy and intensity levels [39,40]. Dendrimers are characterized by being built by segments of molecules giving rise to different generations, that is, macromolecules of different size; those of small size are the best acting as stabilizing agents for NPs coverage, but when they exceed generation 5 (very branched structures) can be more effectively incorporated to the NPs surface due to the great presence of carbonyl and amine functional groups, which facilitate the adsorption of NPs [41,42].

The objective of this work is to obtain a flexible, highly transparent and luminescent film with potential use in optoelectronic devices. For that purpose, CdSe QDs covered by a DAB dendrimer generation 5 with thiol endings functional organic groups (S-DAB_5_) were included in a highly swollen regenerated cellulose film by simple depth-coating process (RC-4/CdSe-QDs@S-DAB_5_ film). Basic information on surface and bulk modification of the QDs nano-engineered film was obtained by FE-SEM and TEM microscopy, XPS and impedance spectroscopy techniques, while their optical properties were determined by photoluminescence spectra as well as by light transmittance/reflectance and spectroscopic ellipsometry measurements. These latter results provide information on film characteristic optical parameters (refraction index and extinction coefficient) and anisotropy due to CdSe-QDs@S-DAB_5_ NPs inclusion in the RC support. This kind of hybrid system with tailoring optical parameters can be of interest in different applications (photovoltaic devices, light-emitting devices, etc.) [19,23,24,25]. 

## 2. Materials and Methods

### 2.1. Preparation of CdSe QDs and Thiolated DAB Generation 5 Dendrimer

The synthesis of CdSe-QDs was performed following the procedure previously indicated in reference [43]. Briefly, CdCl_2_ (0.2 mM) was dissolved in H_2_O (50 mL) and, after total dissolution, 3-mercaptopropyl acid (2.2 mM) was added and left overnight to ensure the coating process, previously to the addition of Se, which was prepared as NaHSe to avoid further oxidation process. CdSe QDs powders with strong fluorescent under UV radiation were obtained after the purification step with EtOH/H_2_O dissolutions. 

Thiolate DAB dendrimer generation 5 (S-DAB_5_) was prepared according to the procedure previously indicated in [44]; 3-mercaptopropanyl-N-hydroxysuccinimide ester, dendritic polyamine DAB-AM and triethylamine in CH_2_Cl_2_ was used for S-DAB_5_ dendrimer synthesis; the obtained dendrimer (formulae: (NHCH_2_CH_2_SH)_64_, mass: 12,808.1323 u.a.m and size of around 6.6 nm) is soluble in aqueous solutions but insoluble in organic solvents. Figure 1a shows a schematic of the S-DAB_5_ dendrimer, while Figure 1b shows the CdSe-QDs@S-DAB_5_ nanoparticles

### 2.2. Preparation of the Cellulosic Films Modified with CdSe-QDs@ S-DAB_5_ Nanoparticles

A highly swelling (>80%), elastic, and transparent film of regenerated cellulose (RC) from Cellophane Española S.A. (Burgos, Spain was selected for easy inclusion of CdSe-QDs@S-DAB_5_ by depth-coating method) [43]. Pieces of the RC-4 film were immersed in an aqueous solution of CdSe-QDs@S-DAB_5_ for 2 h at room temperature, then they were taken off and their surfaces gently dried with paper. These samples will be named RC-4/CdSe-QDs@S-DAB_5_, and a scheme of this flexible and easily handle film is shown in Figure 2. 

### 2.3. Surface Analysis: FE-SEM Microscopy and XPS Spectroscopy

The surface morphology of the cellulosic film coated with CdSe-G_5_ QDs nanoparticles was analyzed using an FEI Talos F200X field-emission scanning electron microscope (FE-SEM) with a double beam (Helios Nanolab 650 de FEI Company, Oxford, UK). Surface chemical characterization of the studied film was carried out by X-ray photoelectron spectroscopy (XPS, Physical Electronics ULVAC-PHI Lake Drive East, Chanhassen, MN, USA). XPS spectra were recorded with a Physical Electronics PHI 5700 spectrometer with X-ray MgK_α_ radiation as the excitation source (300 W, 15 kV, 1253.6 eV). High-resolution spectra were recorded at two take off angles, 45° (standard analysis angle) and 70°, by a concentric hemispherical analyzer operating in the constant pass energy mode at 29.35 eV and using a diameter analysis area of 720 µm. The residual pressure in the analysis chamber was maintained below 10^−9^ Torr during data acquisition. Accurate ±0.1 eV binding energies were determined with respect to the position of the adventitious C 1*s* peak at 284.8 eV. A PHI ACCESS ESCA-V6.0F software package was used for acquisition and data analysis. Atomic concentration percentages (A.C. %) of the sample elements were determined after subtraction of a Shirley-type background considering the corresponding area sensitivity factor for the different measured spectral regions [45].

### 2.4. Optical Characterization

An Edinburgh Instruments FLS920 (Livingston, UK), equipped with a Xe lamp (450 W) as the excitation source and monochromatic LEDs (PicoQuant PLS), controlled by a PDL880-B system, was used for steady-state fluorescence measurements.

Transmittance/reflection measurements were performed with a Varian Cary 5000 spectrophotometer (Agilent Technologies, Santa Clara, CA, USA) provided with an integrating sphere of Spectralon for wavelength ranging between 250–2000 nm. 

Spectroscopic Ellipsometry (SE) measurements were carried out with a spectroscopic ellipsometer (Sopra-Semilab GES-5E) using wavelengths in the range from 200 nm to 1000 nm after striped the back surface of the sample for reduction of interference fringes associated with multiple light reflections at the back interface (see Appendix A). WinElli software v. 2.2 (Sopra-Semilab, Paris, France) was used for data fit. SE is a non-destructive technique that allows the determination of optical/morphological parameters of inorganic and polymeric thin films (refraction index, extinction coefficient or thickness) [46,47,48]. Two characteristic parameters, angles Ψ and Δ, are measured to determine changes associated to film surface or bulk phase [46,49], which are related with differential changes in amplitude and phase between the incident and reflected light waves (through the Fresnel reflection coefficients ratio of polarized light) by [46]: tan(Ψ)e^i^^Δ^ = r_p_/r_s_(1)
where r_s_ and r_p_ indicate the amount of light in perpendicular (s) and parallel (p) planes, as it is schematically shown in Figure 3. Because SE is based on the ratio of two measured values, it is very accurate and reproducible, and no standard sample is required [46]. 

SE measurements were performed at three different incident angles (Φ_ο_ = 65°, 70°and 75°) since it can also give information on sample homogeneity, the presence of surface impurities and roughness [49,50]. 

### 2.5. Impedance Spectroscopy Measurements

Impedance spectroscopy (IS) measurements were performed with dry samples in an electrode/sample/electrode test cell [51]. The electrodes were connected to an Impedance Analyzer (Solartron 1260, Solartron Analytical, Wokingham, UK) and measurements were recorded for 100 data points with frequency (f) ranging between 10 Hz and 10^7^ Hz, at a maximum voltage of 0.01 V. The impedance, Z, is a complex number, Z = Z_real_ + j Z_img_, which can be separated into real and imaginary parts by algebra rules. Electrical parameters (resistance (R) and capacitance (C)) can be determined by analyzing the impedance plot (Z_real_ versus −Z_img_) by considering equivalent circuits. The simplest case, for homogeneous systems, corresponds to a semi-circle and it is due to a parallel association of resistance (R) and capacitor (C) [46], which are related to Z_real_ and Z_img_ by the following expressions:Z_real_ = (R/[1 + (ωRC)^2^])(2)
Z_img_ = − (ωR^2^C/[1 + (ωRC)^2^])(3)
where ω represents the angular frequency (ω = 2πf). However, complex systems usually present distribution of relaxation times and the resulting plot is a depressed semi-circle, which is associated with a non-ideal capacitor or constant phase element (CPE), and its impedance is expressed by [52]: Q(ω) = Y_o_(jω)^−m^, where Y_o_ represents the admittance and m is an experimental parameter (0 ≤ m ≤ 1) and in these cases an equivalent capacitance (C^eq^) can be determined. This kind of analysis allows us to determine the presence/effect of different polymer-modifying elements [53,54]. 

## 3. Results and Discussion

### 3.1. Surface and Bulk Analysis

Figure 4 shows the FE-SEM image of the RC-4/CdSe QDs-DAB_5_ film, where the presence of the CdSe QDs@S-DAB_5_ nanoparticles on the surface of the RC-4 film is clearly observed, while the EDAX analysis gives the following values, in atomic (%): C (29.5%); O (37.3%); S (21.4%); Cd (1.9%) and Se (0.2%). In addition, TEM images of the synthesized nanoparticles (in solution) and the EDAX analysis (performed with a Philips CM 200 microscope) [43] are presented as Appendix A (Appendix A and Appendix A, respectively).

On the other hand, when the surface analysis is carried out by XPS, it is possible to detect the presence of other atoms such as nitrogen, by the N 1*s* signal from the dendritic structure. Here the surface chemical concentrations (in A.C.%) at the standard analysis angle (45°) are: C 1*s* (77.5%); O 1*s* (20.5%); N 1*s* (0.4%); Cd 3*d* (0.02%) and Se 5*d* (0.02%); moreover, a slight increase in Cd 3*d* and Se 5*d* values (0.03%) was obtained from measurement performed at a higher incident angle (70°), which corresponds to deeper analysis, which seems to confirm the presence of CdSe QDs-DAB_5_ into the film structure (A.C. % of the different elements at 70° incident angle are indicated in Appendix A. 

The XPS study of the surface provides the chemical oxidation state of the characteristic QDs ions (Figure 5). The analysis of the C 1*s* reveals the presence of three main contributions assigned to: (i) adventitious carbon (*sp*^2^ hybridized carbon) –C-C–/–C-H at 284.8 (69.53%); (ii) –C-OH/–C-N at 286.39 (24.1%) and (iii) –C=O at 288.2 eV (6.5%) functional groups (Figure 5a), related mainly to the presence of dendrimer molecule [55]. On the other hand, the N 1*s* spectra (Figure 5b) exhibits a major peak at around 400.0 eV and it was attributed to N atoms bonded with *sp*^2^ -hybridized C atoms [56]; Figure 5b also shows the Cd 3*d* signal from the QDs, at 405.1 and 412.1 eV, assigned to the 3*d*_3/2_ and 3 *d*_5/2_ orbitals. Figure 5c shows the O 1*s* signal, a peak at 532.5 eV, assigned to the C–O and C-O-C signal. It should be indicated that the Se signal was not observed due to its low sensitivity factor and dendrimer coverage.

Another technique able to give information on bulk material and interface (electrode/sample surface) contributions is impedance spectroscopy (IS). IS is an alternating current (a.c.) technique commonly used for electrical characterization of homogeneous and heterogeneous materials as well as for composite systems (such as electrolyte/sample commonly named Electrochemical Impedance Spectroscopy or EIS) using equivalent circuits as models [53,54,57,58]. IS provides quantitative and/or qualitative information related to charge movement/adsorption for the analyzed samples by means of the electrical resistance or capacitance (equivalent capacitance for non-homogeneous systems, C^eq^ = (RY_o_)^(1/m)^/R [52]) respectively, which can be significantly affected by the structure of the analyzed system and material characteristics. Figure 6a shows the Nyquist plot (Z_real_ vs. −Z_img_) obtained for dry samples of the original RC-4 film and the modified RC-4/CdSe-QDs@S-DAB_5_ one, where differences in both bulk sample and electrode/sample interface can be observed. For comparison reason, the Nyquist plot for another nano-engineered film, RC-4/CdSe-QDs@S-DAB_2_, modified with CdSe QDs covered by S-DAB_2_ dendrimer (formula and mass: NHCH_2_CH_2_SH)_8_ and 1478.3187 u.m.a.) is shown in Figure 6b, and in this case, practically no differences were obtained for the electrode/sample region, although slight differences in the bulk films contribution can be observed. Figure 6c,d show the Bode plots (Z_real_ vs. f and −Z_img_ vs. f, respectively) which permit seeing in a clearer way the differences between RC-4 and RC-4/CdSe-QDs@S-DAB_5_ films, where the slight shift to lower frequency obtained for this latter film is an indication of its more compact structure associated with the nanoparticles inclusion. 

The fit of the two slightly depressed semicircles shown in Figure 6a by a non-linear program allows us the estimation of the electrical resistance (R) and equivalent capacitance (C^eq^) of the films, which permits the determination of their electrical conductivity (σ = d/S∙R) and dielectric constant (ε = C∙d/S), where d and S represent the thickness and surface of the films, respectively. The values obtained for the RC-4/CdSe-QDs@S-DAB_5_ film are: σ = 3.1 × 10^−7^ (Ω∙m)^−1^ and ε = 8.1, which represent a reduction of around 18% in the conductivity and an increase of 7% in the dielectric constant with respect to the support film. Therefore, IS results seem to confirm the presence of CdSe-QDs@S-DAB_5_ into the structure of the cellulosic film as well as on its surface. 

### 3.2. Optical Analysis

Optical characterization techniques, photoluminescence (PL), light transmission and spectroscopic ellipsometry (SE) are of great interest in the analysis of thin films due to their non-invasive/non-destructive character, providing important characteristics of the analyzed samples. 

One of the most significant properties of CdSe QDs is their fluorescent emission, with a maximum intensity of 534 nm; consequently, the luminescence character of the RC-4/CdSe-QDs@S-DAB_5_ film shown in Figure 7, obtained at an excitation source of 475 nm, was expected. The emission spectra of the QDs-dendrimer modified film show an intensity maximum at 536 nm and exhibits higher intensity than that presented by the CdSe QDs. This fact is associated with the presence of the dendrimer on the surface of the QDs, since the thiol terminal functional groups increase the QDs emission process, as was already indicated in a previous study [59], but they do not change the emission wavelength. 

Spectroscopic ellipsometry (SE) results are presented in Figure 8. In particular, Figure 8a,b show a comparison of the wavelength dependence of the experimental parameters, tan(Ψ) and cos(Δ), measured at different light incident angles (Φ_o_ = 65°, 70° or 75°) for the RC-4/CdSe-QDs@S-DAB_5_ film and the RC-4 support, where differences depending on both incident angle and film can be observed. Optical characteristic parameters such as the refractive index (n) and the extinction coefficient (k) can be determined from SE measurements using the ellipsometer software, and their dependence on wavelength at the different light incident angles is shown in Figure 8c,d, respectively, for both films. 

As it can be observed in Figure 8c rather similar values for both refraction index and extinction coefficient were obtained at the different light incident angles for the RC-4 support, which is an indication of film smooth surface and homogeneity, and the small differences are attributed to surface impurities associated to environmental contamination; the following average values were determined: <n> = 1.56 ± 0.03 and <k> = 0.06 ± 0.03, which do not differ significantly from those reported for regenerated cellulose (1.48) and other polymers (between 1.53 and 1.58) measured at a unique wavelength of 589 nm [60], and zero in the case of extinction coefficient (no light absorption). However, higher values variability depending on the light incident angle was obtained for the RC-4/CdSe-QDs@S-DAB_5_ film, which is attributed to sample inhomogeneity caused by inclusion of the CdSe-QDs@S-DAB_5_ nanoparticles, as well as the surface roughness associated to CdSe QDs presence already established by SEM images. The following average values for the refraction index were estimated: <n(65°)> = 1.58 ± 0.05, <n(70°)> = 1.62 ± 0.03 and <n(75°)> = 1.59 ± 0.03; in the case of the extinction coefficient: <k(65°)> = 0.32 ± 0.03, <k(70°)> = 0.27 ± 0.09 and <k(75°)> = 0.22 ± 0.11, indicating higher light absorption by the modified film. Consequently, the increase of n and k values determined for the RC-4/CdSe-QDs@S-DAB_5_ film with respect to the RC-4 support is an indication of CdSe QDs contribution in the optical behavior of the modified film (reported values for CdSe refractive index range between 2.35 and 2.64 [61]). 

The effect of the type of QDs (bare silicon dots, carbon dots, or nitrogen-doped carbon dots) on optical characteristics of the RC-QDs modified films was already studied in a previous paper [28] by comparing wavelength dependence of n and k parameters, and differences in both curve shape and values depending on the selected QDs nanoparticles were obtained. Therefore, the effect on optical parameters associated with dendrimers coverage of the CdSe QDs was considered in this work; then, three different generations of S-DAB dendrimers (2, 3 and 5 generations) were considered: (i) the RC-4/CdSe-QDs@S-DAB_5_ film, (ii) the RC-4/CdSe-QDs@S-DAB_2_ film (CdSe QDs covered with the generation 2 S-DAB dendrimer indicated above), and (iii) the RC-4/CdSe-QDs@S-DAB_3_ film (CdSe QDs covered with the generation 3 dendrimer; formulae and mass: (NHCH_2_CH_2_SH)_16_ and 3096.8635 u.a.m.), and the wavelength dependence of n and k for these three RC-4/CdSe-QDs@S-DAB_x_ modified films, at a light incident angle of 70°, is shown in Figure 9. As it can be observed, n and k parameters determined for the three films exhibit a similar type of wavelength dependence, but differences in the corresponding values depending on dendrimer generation were obtained, with the following sequence: RC-4/CdSe-QDs@S-DAB_5_ film > RC-4/CdSe-QDs@S-DAB_3_ film > RC-4/CdSe-QDs@S-DAB_2_, being this fact more significant for the extinction coefficient. Consequently, the choice of both the type of QDs and their coating allows us to obtain films with differentiated optical properties. On the other hand, it should be indicated that SE results also provide information on the real and the imaginary parts of the dielectric constant (ε_r_ and ε_i_, respectively) considering that ε = (n + i k)^2^ [46], which could be of interest in some applications. 

The effect of CdSe-QDs inclusion on light transmission and reflection characteristics of the support film was also investigated. Figure 10a shows the transmittance spectra for the RC-4 and the RC-4/CdSe-QDs@S-DAB_5_ films, and practically no effect due to the presence of the nanoparticles is observed in the visible region, but slight differences in the near-infrared (NIR) region were obtained (1% average reduction); this fact seems to be related to the dendrimer coverage since similar dependence for the whole wavelength range was obtained for the RC-4/CdSe-QDs@S-DAB_5_ and the RC-4/CdSe-QDs@S-DAB_3_ films (see the insert in Figure 10a). High transmittance values are of interest for optical applications; in fact, the value obtained (>90%) is like that reported for carboxymethyl cellulose films modified with carbon dots with application in agriculture (sunlight conversion) [21].

The comparison of light reflectance spectra obtained for the RC-4 and the RC-4/CdSe-QDs@S-DAB_5_ films presented in Figure 10b also shows similar dependence, but an average reflectance % reduction in the NIR region of ~5% due to QDs inclusion. However, the type of dendrimer seems to affect light refection since a more significant reduction (around 20% for the whole range of wavelength) was obtained for the RC-4/CdSe-QDs@S-DAB_3_ film.

## 4. Conclusions

A photoluminescence film was obtained by the inclusion of CdSe QDs covered by thiolate DAB_5_ dendrimer nanoparticles (CdSe-QDs@S-DAB_5_) into a highly swollen and elastic regenerated cellulose (RC) film support by depth-coating into an aqueous solution of these nanoparticles. The nano-engineering RC-4/CdSe-QDs@S-DAB_5_ film exhibits modified optical characteristics when compared with the RC-4 support, with a significant increase in the values of the extinction coefficient but only around 4% in the refraction index. Moreover, the influence of the cover-dendrimer generation (S-DAB_2_, S-DAB_3_ or S-DAB_5_) on n and k parameters was also demonstrated, indicating that both the types of QDs and the coating (in case) influence both optical parameters. The RC-4/CdSe-QDs@S-DAB_5_ film also shows very high transmittance (>90%), with only small changes at the near-infrared region for both light transmission and reflection (around 3%), when compared with the support RC-4 film, and only small changes associated with the dendrimer generation were obtained.

## Figures and Tables

**Figure 1 nanomaterials-12-00484-f001:**
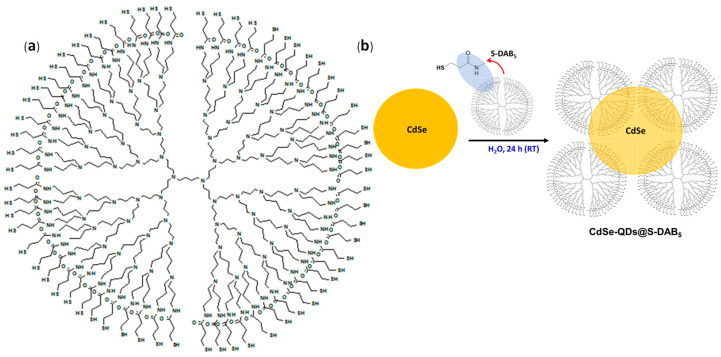
Scheme (**a**) S-DAB_5_ and (**b**) CdSe-QDs nanoparticles coated by S-DAB_5_ dendrimer (ChemWindow Chemical Structure Drawing Software from Wiley).

**Figure 2 nanomaterials-12-00484-f002:**
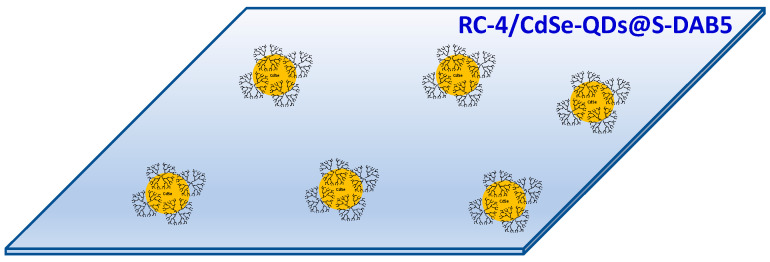
Scheme of the RC-4/CdSe-QDs@S-DAB_5_ film.

**Figure 3 nanomaterials-12-00484-f003:**
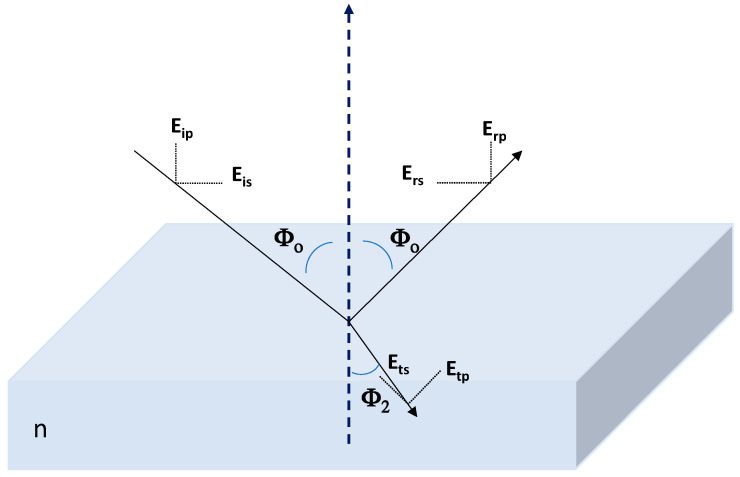
Scheme of spectroscopy ellipsometry measurement for a homogeneous sample. Φ_o_ incident angle; Φ_2_: refraction angle. E_is_: perpendicular component of the incident electric field; E_ip_: parallel component of the incident electric field; E_rs_: perpendicular component of the reflected electric field; E_rp_: parallel component of the reflected electric field; E_ts_: perpendicular component of the refracted electric field; E_tp_: parallel component of the refracted electric field. Sample refraction index: n.

**Figure 4 nanomaterials-12-00484-f004:**
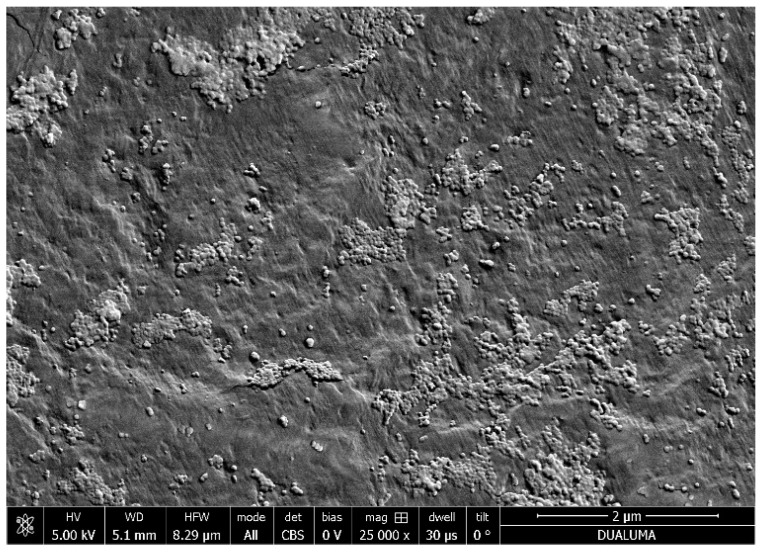
FE-SEM images of RC-4/CdSe-QDs@S-DAB_5_ film.

**Figure 5 nanomaterials-12-00484-f005:**
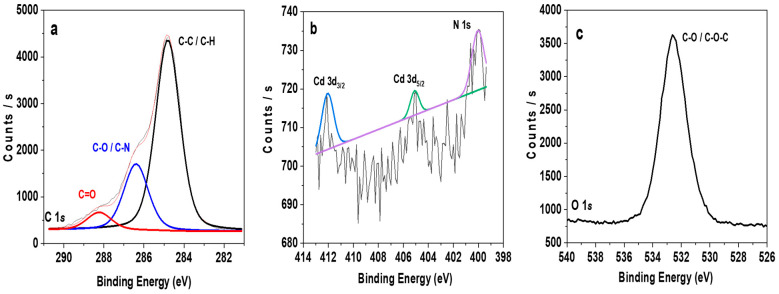
XPS core level spectra of RC-4/CdSe QDs-DAB_5_ film: (**a**) C 1*s*; (**b**) Cd 3*d* and N 1*s*; (**c**) O 1*s*.

**Figure 6 nanomaterials-12-00484-f006:**
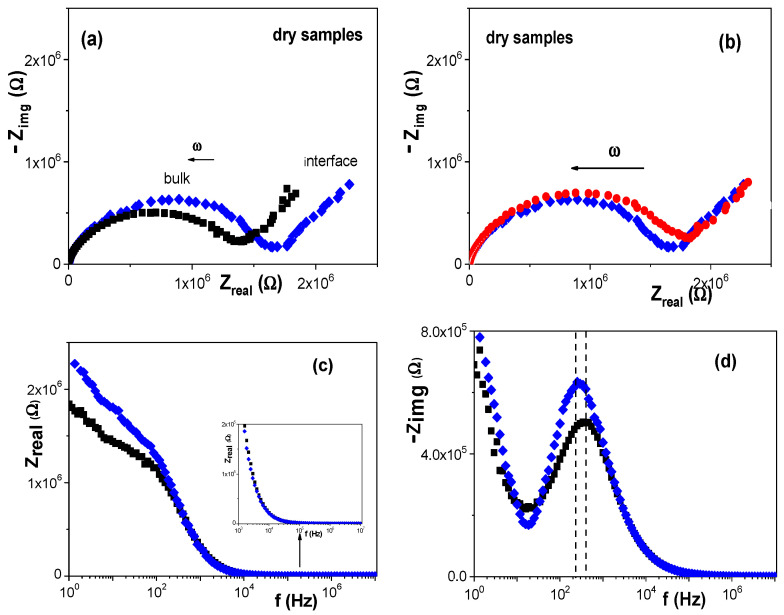
Impedance plots: (**a**) and (**b**) Nyquist plot ((Z_real_ vs. −Z_img_); (**c**) Z_real_ as a function of frequency; (**d**) −Z_img_ as a function of frequency. (■) RC-4 film; (♦) RC-4/CdSe-QDs@S-DAB_5_ film; (●) RC-4/CdSe-QDs@S-DAB_2_ film.

**Figure 7 nanomaterials-12-00484-f007:**
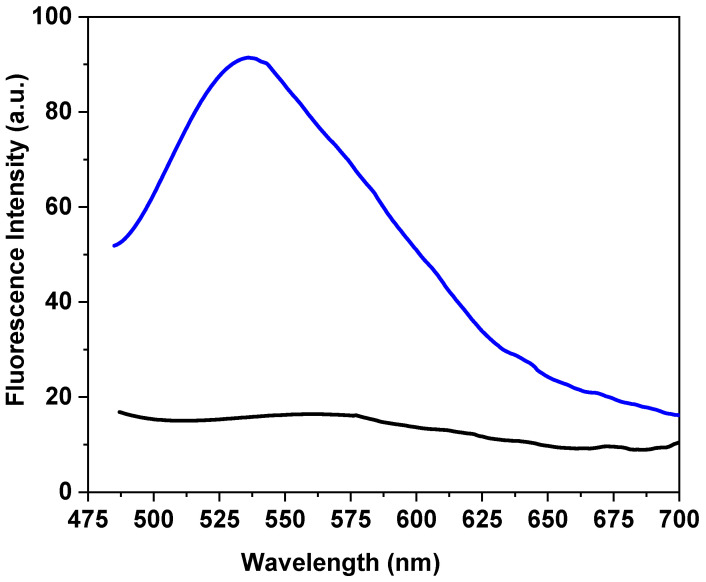
Comparison of the fluorescence spectra for the RC-4/CdSe-QDs@S-DAB_5_ (blue line) and RC-4 (black line) films excited at 475 nm.

**Figure 8 nanomaterials-12-00484-f008:**
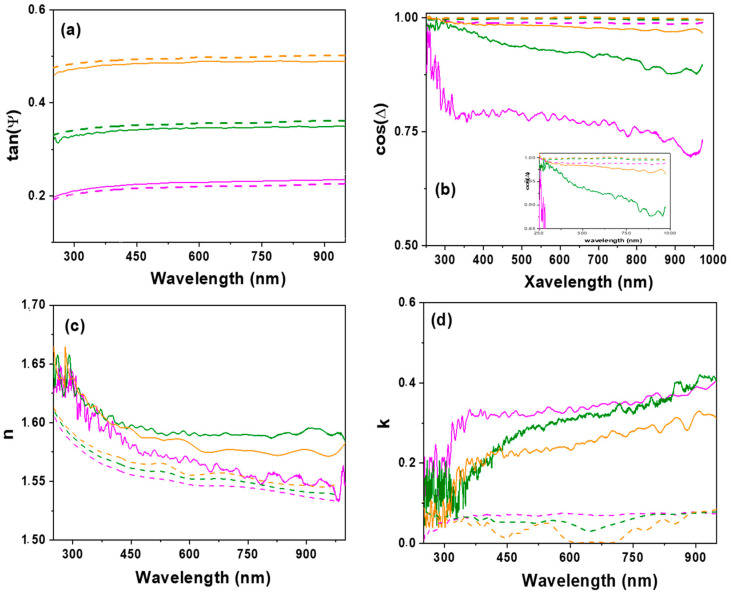
Wavelength dependence of: (**a**) tan (Ψ); (**b**) cos(Δ); (**c**) refraction index; (**d**) extinction coefficient, for the RC-4 support (dashed lines) and the RC-4/CdSe-QDs@S-DAB_5_ film (solid lines) determined at different light incident angles: 65° (magenta line), 70° (green line) and 75° (orange line).

**Figure 9 nanomaterials-12-00484-f009:**
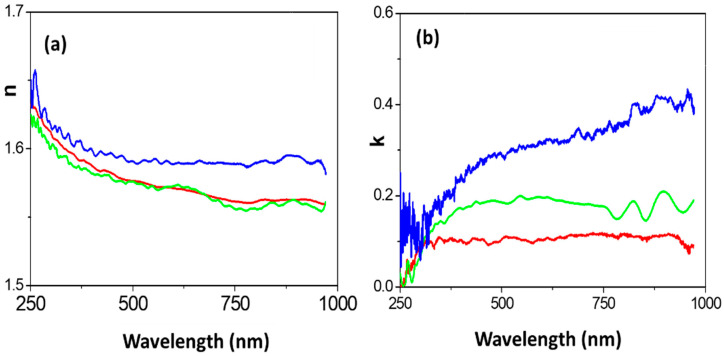
Wavelength dependence of refraction index (**a**) and extinction coefficient (**b**) for: RC-4/CdSe-QDs@S-DAB_5_ film (blue line), RC-4/CdSe-QDs@S-DAB_3_ film (green line) and RC-4/CdSe-QDs@S-DAB_2_ film (red line) at 70° light incident angle.

**Figure 10 nanomaterials-12-00484-f010:**
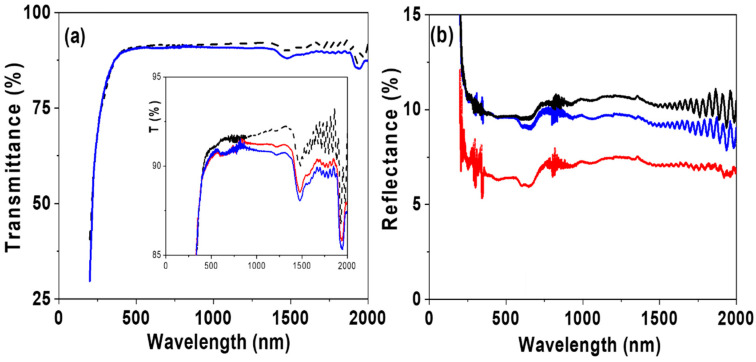
Wavelength dependence of: (**a**) light transmittance and (**b**) light reflection. RC-4 film (black line), RC-4/CdSe-QDs@S-DAB_5_ film (blue line) and RC-4/CdSe-QDs@S-DAB_3_ film (green line).

## Data Availability

Not applicable.

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
