# Peer review of "Optical and Physicochemical Characterizations of a Cellulosic/CdSe-QDs@S-DAB5 Film"

_nanomaterials, 2022, doi:10.3390/nano12030484_

Round 1
Reviewer 1 Report
The impedance spectroscopy measurements show a difference between the RC-4 film and the RC-4/ CdSe-QDs@S-DAB5 film. Could the authors explain how the reduction of the conductivity and the increase of the dielectric constant could indicate the presence of CdSe-QDs@S-DAB5 and not of something else?
To avoid any confusion, in caption of Figure 6, could the authors specify which curve corresponds to which film ?
Author Response
Attached are the answers requested

Reviewer 2 Report
In this work, a cellulosic/CdSe-QDs@S-DAB5 film were prepared by Algarra et al., and its optical properties were tested by the different spectroscopy technology. This work is interesting in the field of optical materials. However, there are some problems to be solved before considering it.
- In Figure 3, the TEM images of the samples are not clear, the authors should provide the images with higher resolution ratio. The same questions can be found in Figure 5(a)(b) and (c)
- What is the fluorescence quantum yield of the sample? The author should test this property of the sample.
- What is the different of the optical properties of QDs in the solvent and on the film?
- What is the software used when the author prepared the Figure 1? Please add this introduce in Materials and Methods.
- There maybe exist some errors in Figure 8, the author should check and revise it.
Author Response
Attached the answers requested

Reviewer 3 Report
The authors reported the manuscript entitled "Optical characterization of a cellulosic/CdSe-QDs@S-DAB5 film" in a very organized way but the paper may be accepted after major revision. Queries attached.

Author Response
Attached are the answers requested

Reviewer 4 Report
The manuscript presents the development of a cellulosic/CdSe-QDs@S-DAB5 film, with an interesting composition. The work is relevant. Also, it is in the scope of the journal. However, there are some issues that must be clarified. Additionally, the authors should refer to the real importance of applying the film, in a more objective way.
- Title – Authors should give an appropriate title, as it does not cover the entire work. More than the optical characterization, physicochemical characterization of the material was also carried out.
- Introduction – In the introduction, the authors should reinforce the importance of the film they have developed and specifically indicate applications where the film can bring advantages.
- Line 60 to 62 – Please provide synthetic explanation.
What authors mean with “synthesized with very high precision”? and “making of them an attractive platform for very diverse applications”.
- Lines 85, 86 – Please specify “different applications” or provide precise information.
- Materials and methods
- Lines 103 to 105 – the described information should be moved for the Results section
- Lines 124 to 128 – Are both TEM and SEM performed with the same equipment? Authors should write this information more explicitly.
- In Figure 3(a) it is possible to observe, in addition to the black dots, a reticulated tubular structure throughout the image. Please indicate what this structure might be and give a brief description of what you observe.
Results and Discussion
- In impedance spectroscopy (IS) study, the authors indicate that electrode/sample/electrode test cell was used. However, they do not mention which solution was used to carry out the measurements. Provide additional information on this topic.
- In the RC-4/CdSe-QDs@S-DAB5 film formation, are CdSe-QDs@S-DAB5 typically adhered to both sides of the film? If it's on only one side, how do you select the side? If it's on both sides, how do you know you don't have too much amount?
- The resolution of figure 5 must be improved
- Figure 6 should include the color legend and respective material under study
- Legend of Figure 8: the authors mention "film RC-4 / CdTe-QDs @ S-DAB3 (purple line)". I think you should correct for “red line”.
- Supplementary Information - the resolution of Table SI.2. A and Figure SI.2. must be improved.
English changes required (examples):
Line 235 – “slight sift to lower” -> should be shift?
Line 375 – “into a aqueous solution” -> “an aqueous”
Author Response
Attached are the answers requested

Round 2
Reviewer 2 Report
The manuscript has been revised and improved compared with the first round version. Now there is some problem with picture order (such as there are two pictures both named after Figure 2). Suggested authors to check the picture order and the corresponding discussion in the manuscript. At the same time, clearer images need to be provided for Figure 3. FE-SEM and Figure 2: Scheme of spectroscopy ellipsometry measurement because they seem to be not clear in current PDF version.
Author Response
Dear Reviewer
Attached is the corrections

Reviewer 3 Report
Authors have revised the manuscript in the light of referees comments very well but two figures have mentioned Fig.2.Please correct it. Please provide clear picture of SEM.
Paper may be accepted for publication with minor changes.
Author Response
Attached is the requested answered points
Thanks
